# Frequency of MTB and rifampicin resistance MTB using Xpert-MTB/RIF assay among adult presumptive tuberculosis patients in Tigray, Northern Ethiopia: A cross sectional study

**Araya Gebreyesus Wasihun**[1]*, **Tsehaye Asmelash Dejene**[1,2], **Genet Gebrehiwet Hailu**[1]

**1** Department of Medical Microbiology and Immunology, School of Medicine, Mekelle University, Mekelle, Ethiopia, **2** Department of Medical Microbiology and Immunology, School of Medicine, Aksum University, Aksum, Ethiopia

* araya13e25@gmail.com

**Data Availability Statement:** All relevant data are within the manuscript.

## Abstract

### Background

Multidrug-resistant tuberculosis (MDR-TB) continues to be a global health problem. Data on rifampicin resistance MTB using Xpert- MTB/RIF assay in Ethiopia, particularly in the study area is limited. The aim of this study was to determine the frequency of MTB and rifampicin resistant-MTB among presumptive tuberculosis patients in Tigray, Northern Ethiopia.

### Methods

A multicenter retrospective study was conducted among presumptive TB patients from five governmental hospitals and one comprehensive specialized teaching hospital in Tigray regional state. Records of sputum sample results of presumptive MTB patients with Xpert-MTB/RIF assay from January 2016 to December 2019 were investigated. Data extraction tool was used to collect data from registration books and analyzed using SPSS ver.21 statistical software. Statistical significance was set at p-value ≤ 0.05.

### Results

Of the 30,935 presumptive adult TB patients who have provided specimens for TB diagnosis from January 2016 to December 2019, 30,300 (98%) had complete data and were included in this study. More than half, 17,471 (57.7%) were males, and the age of the patients ranged from 18–112 years, with a median age of 40.65 (interquartile 29.4–56.5 years). Majority, 28,996 (95.7%) of the participants were treatment naïve, and 23,965 (79.1%) were with unknown HIV status. The overall frequency of MTB was 2,387 (7.9% (95% CI: 7.6–8.2%); of these, 215 (9% (95% CI: 7.9–10.2%) were rifampicin resistant-MTB. Age (18–29 years), HIV positive and previous TB treatment history were significantly associated with high MTB (p < 0.001), whereas gender (being female) was associated with low MTB (p < 0.001). Likewise, rifampicin resistant-MTB was more prevalent among relapse (p < 0.001) and failure

**Funding:** No funding was obtained from any funding organizer to run the research.

**Competing interests:** The authors have declared that no competing interests exist.

**Abbreviations:** HIV, Human immunodeficiency virus; MDR-TB, Multidrug resistant tuberculosis; MTB, Mycobacterium tuberculosis; RR, Rifampicin resistant tuberculosis; TB, Tuberculosis; WHO, World Health Organization; DOTS, Directly observed treatment, short-course.

cases (p = 0.025); while age group 30–39 years was significantly associated with lower frequency of rifampicin resistant-MTB (p = 0.008).

## Conclusion

Frequency of MTB among tuberculosis presumptive patients was low; however, the problem of rifampicin resistant-MTB among the tuberculosis confirmed patients was high. The high frequency of MTB and RR-MTB among previously treated and HIV positive patients highlights the need for more efforts in TB treatment and monitoring program in the study area.

## Introduction

Tuberculosis (TB) is one the top ten causes of mortality and the first killer among infectious diseases worldwide. Multidrug resistant *Mycobacterium tuberculosis* (MDR-MTB), defined as resistant to at least isoniazid and rifampicin, is a major global health problem. According to the 2019 report of the WHO, globally, an estimated 10.0 million (range, 9.0–11.1 million) people fell ill with TB and about 1.2 million (range, 1.1–1.3 million) TB deaths among HIV-negative people were reported in 2018. Similarly, there were about half a million new cases of rifampicin-resistant TB (of which 78% had multidrug-resistant TB) in the 2018 [1]. Delay in early diagnosis and appropriate treatment initiation, and high prevalence of HIV in resource limited settings made TB and MDR-TB associated morbidity and mortality to be quite high [2, 3].

The WHO endorsed Xpert MTB/RIF assay in 2010, an automated molecular system which detects both DNA of MTB and rifampicin resistance (RR) simultaneously [4]. Rifampicin resistant -MTB (RR-MTB) is a proxy marker for MDR-TB in more than 90% of the cases [5]. Initially, the assay was indicated for patients with TB/HIV co-infection, presumptive MDR-TB and paediatrics TB patients [6]. Three years after its implementation it was recommended for all TB presumptive patients [7]. In Ethiopia Xpert- MTB/RIF assay was implemented in all general and referral hospitals since 2014 [8].

Ethiopia is one of the 30 high TB, TB/HIV and MDR-TB burdened countries with a rank of 15th among the high MDR-TB countries with more than 5800 estimated MDR-TB cases each year [9]. A systematic review from Ethiopia reported that 2.18% of TB treatment naïve and 21.07% of previously treated patients had MDR-TB nationwide [10]. Most studies in Ethiopia used the conventional culture and sensitivity methods not the automated Xpert -MTB/RIF assay. There are few studies in Ethiopia such as: Addis Ababa [11], Amhara regional state [4] and Southern Ethiopia [12] on prevalence of TB and rifampicin resistant MTB (RR-MTB) using Xpert- MTB/RIF assay. However, these studies were far from complete because of their area coverage. For example, the report from the Amhara regional state and south Ethiopia included single hospital each and used small sample size. Similarly, the study conducted in Addis Ababa included 12,414 samples from four health facilities, but this cannot represent the national level prevalence. The limitations of the previous studies calls for more data to be generated from each region with representative sample to forward reasonable findings and recommendations to help policy makers and implementers to plan and design proper intervention strategies to control TB.

In Tigray regional state, a total of 9,594 TB cases were reported in 2015 [13]. There are only two studies on MDR-TB in this region [14, 15]. Both of the studies, however, were done on presumptive MDR-TB patients (failure, relapsed, and who have contact with MDR TB

patients) which could not show the magnitude of MTB and RR-MTB among the presumptive TB patients in the region. Hence, addressing this knowledge gap on the prevalence and associated factors of TB and RR-MTB among presumptive adult TB patients in the study area is rational to help policy makers and implementers to plan and design proper intervention to achieve the strategy "End TB by 2035."

## Methods

### Study design and study population

**Study setting.**   Tigray Regional state is the North most region of the Federal Democratic Republic of Ethiopia with population size of 6,960,003 with an area of 54, 572.6 km$^2$. The capital city of Tigray is Mekelle, located 783 km north of Addis Ababa, the capital of Ethiopia. The region is bordered by Sudan in the west, Eritrea in the north, Afar regional state in the East and Amhara regional state in the South. It is administratively divided into seven Zones and 52 districts (34 rural and 18 urban). In the region, health services are provided by one teaching and specialized hospital, 12 general hospitals, 22 primary hospitals, 204 health centers, 712 health posts [village clinic] and 500 private health facilities. In the region Xpert-MTB/RIF assay for TB diagnosis is given in the general hospitals not the primary hospitals.

A multicenter health facility based retrospective cross sectional study design was used to collect data from January 2016 to December 2019 from five governmental general hospitals. In this study, general hospitals which introduced Xpert since 2016 were included. Thus, from the 10 general hospitals which introduced Xpert-MTB/RIF assay for MTB diagnosis five hospitals, and one comprehensive specialized teaching hospital were included namely: Adigrat, Wukro, Mekelle, Lemelem Karli, Alamata and Ayider comprehensive specialized teaching hospital located in the three zones of the region (Eastern zone, Southern zone and Mekelle special zone. Directly observed treatment, short-course (DOTS) TB treatment services are given in all the health facilities. The region had three MDR-TB treatment initiation centers and 52 treatment follow-up centers (**Fig 1**).

The source population were all adult patients with clinical signs and symptoms suggestive of MTB and visited the hospitals between January 2016 and December 2019, and gave sputum samples for Xpert MTB/RIF assay. Our study participants were all adult patients (≥18 years) whose data of age, sex, Xpert MTB/RIF results, HIV status and MTB treatment history were recorded in the registration book. Whereas, those children and with any missing information in age, gender, Xpert MTB/RIF results, invalid, indeterminate Xpert MTB/RIF results, HIV status and TB treatment history were excluded from the study.

**Inclusion criteria.**   We included all presumptive TB adult patients (above ≥18 years) with complete record of age, sex, Xpert -MTB/RIF results, HIV status, and TB treatment history. Whereas, children and adults with any missing record on age, gender, Xpert- MTB/RIF results, invalid, indeterminate Xpert MTB/RIF results, HIV status, and TB treatment history were excluded from the study.

### Variables

**Outcome variable.**   MTB and RR-MTB among presumptive TB patients.
**Independent variables.**   Age, gender, HIV status and TB treatment history.

### Operational definitions

**New cases.**   Patients have never been treated for TB before

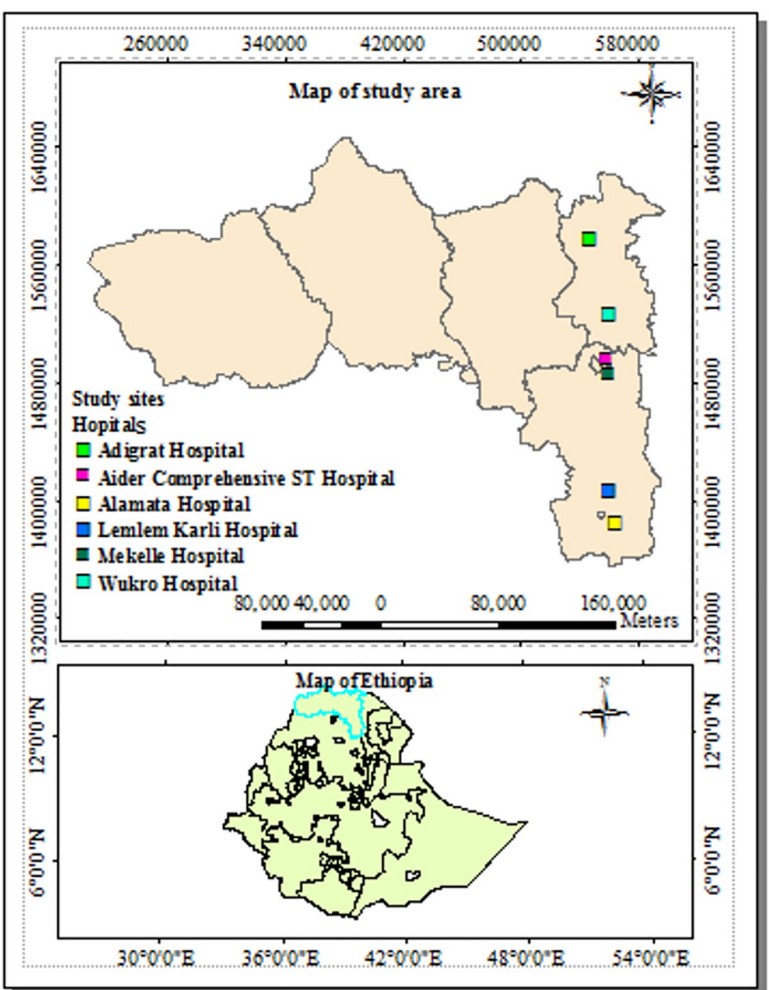

**Fig 1. Map of the study area.**

**Relapse case.** Is a TB patient who has become (and remained) culture negative while receiving therapy but after completion of therapy becomes: culture positive again

**Lost to follow up.** A TB patient who did not start treatment or whose treatment was interrupted for 2 consecutive months or more

**Failure case.** Is a TB patient whose sputum smear or culture is positive at month 5 or later during treatment

**MDR-TB.** TB that does not respond to at least isoniazid and rifampicin, the most important first-line anti-TB drugs

**Rifampicin-resistant TB (RR-TB).** Defined as resistance to rifampicin detected using genotypic or phenotypic methods with or without resistance to other first-line anti-TB drugs

## Data collection

Patients' socio-demographic characteristics (such as age and sex) and clinical-related data (such as Xpert MTB/RIF results, HIV status, and MTB treatment history) were collected using a structured data extraction sheet from Xpert MTB/RIF paper based registration books in each health facilities.

## Laboratory processing

A single sputum sample per patient was used for the diagnosis of TB using Xpert-MTB/RIF assay (Cepheid, Sunnyvale, CA, USA). Briefly, after sputum was collected, it was mixed with sample reagent buffer in 1:2 (sample: sample reagent buffer) volume ratio. Then, closing it tightly, vortexed for 15 seconds and allowed to stand at room temperature for 10 min. It was again vortexed after 10 min and allowed to stand for 5 min, using the Pasteur pipette provided with the kit >2mL of the (just above 2mL mark on pipette) processed sample was put into the Xpert -MTB/RIF cartridge. Then the cartridge with the specimen was loaded to the Xpert machine. Finally, results were collected from the Xpert computer after 2h [16].

## HIV testing

Rapid HIV test was done according to the national algorithm of the Federal Ministry of Health of Ethiopia.

## Data analysis

After data completeness is checked, data was entered and analyzed using SPSS Version 21. Frequency, mean, range and standard deviation were computed. Chi-square and logistic regression analysis were done to identify the associated factors with MTB and RR-MTB. Significant variables in binary logistic regression were analyzed using multiple logistic regressions to identify variables which had association with MTB and RR-MTB at $p \leq 0.05$.

## Ethical consideration

Before the study was conducted, ethical clearance was obtained from Aksum University; College of Health Sciences Institutional Review Board (IRB). Besides, a letter of cooperation was written from the Tigray Regional Health Bureau to each study hospitals and permission was obtained accordingly.

## Result

### Socio-demographic, clinical characteristics and MTB results

Out of the total 30, 935 presumptive adult TB patients who have provided sputum samples for MTB diagnosis, 30,300 (98%) had complete data and were included in this study. More than half 17,471 (57.7%) were males and the median age was 40.65 (interquartile 29.4–56.5 years). The majority, 28,996 (95.7%) and 23,965 (79.1%) of the participants were treatment naïve and with unknown HIV status, respectively. Overall, the frequency of MTB was 2387(7.9%), of those, frequency of RR-MTB was 215 (9%) (**Table 1**).

### Associated risk factors of MTB infections

In this study, females were 14% times less likely [Adjusted Odds Ratio (AOR) = 0.86; 95% CI = 0.79, 0.94, $p < 0.001$] to have MTB compared to males. The odds of having MTB showed a decreasing trend by age. Patients whose age was greater than 29 years were less likely to have MTB compared to 18–29 years [$p < 0.001$]. Likewise, HIV positive patients were 1.54 times [AOR: 1.54; 95% CI: 1.33–1.72, $p < 0.001$]. The odds of MTB was higher among previously treated patients [$p < 0.001$] [**Table 2**].

**Table 1. Socio-demographic, clinical characteristics and MTB result among presumptive adult patients in Eastern, Mekelle and Southern Zones of Tigray, Ethiopia, 2016–2019 (N = 30,300).**

| Variables | Frequency | % |
|---|---|---|
| **Gender** | | |
| Male | 17471 | 57.7 |
| Female | 12829 | 42.3 |
| **Age** | | |
| 18–29 | 7453 | 24.6 |
| 30–39 | 6300 | 20.8 |
| 40–49 | 5349 | 17.7 |
| 50–59 | 4337 | 14.3 |
| 60–69 | 3749 | 12.4 |
| 70–112 | 3112 | 10.3 |
| **HIV Status** | | |
| Positive | 2675 | 8.8 |
| Negative | 3660 | 12.1 |
| Unknown | 23965 | 79.1 |
| **TB Treatment History** | | |
| New case | 28996 | 95.7 |
| Relapse | 1222 | 4 |
| Lost | 25 | 0.1 |
| Failure | 57 | 0.2 |
| **MTB Result** | | |
| Detected | 2387 | 7.9 |
| Not detected | 27913 | 92.1 |
| **RR_MTB Result (N = 2,387)** | | |
| RR _MTB detected | 215 | 9 |
| RR _MTB not detected | 2172 | 91 |

RR = rifampicin resistance.

## Associated risk factors of RR-MTB

Of the total 2,387 MTB confirmed patients, 215 (9%) of them were infected by RR-MTB. Patients whose age was between 30–39 years were 49% times less likely to have RR-MTB [AOR = 0.51; 95% CI = 0.31, 0.84, p = 0.008] compared to the age groups of 18–29 years. On the other hand, RR-MTB was significantly prevalent among relapse cases [AOR = 3.26; 95% CI = 2.14, 4.97, p < 0.001] and failure cases [AOR = 3.75; 95% CI = 1.18, 11.92, p = 0.025] [Table 3].

## Frequency of MTB and RR- MTB by study years

Fig 2 compares the frequency of MTB and RR-MTB by study years. The number of MTB suspected patients who visited the hospitals increased from 3281 in 2016 to 11023 in 2018. Similarly, the absolute number of MTB positive has also increased from 408 in 2016 to 793 in 2018. However, the actual of percent of MTB frequency significantly decreased from (12.4%) in 2016 to (6.8%) in 2019 (p < 0.001, data not shown). Similarly though the absolute number of RR-MTB positive cases shows an increment from 42 in 2016 to 65 in 2018, it was not statistically significant (p > 0.05) [Fig 2].

**Table 2. Frequency of MTB by gender, age, treatment history, and HIV status in Eastern, Mekelle and Southern Zones of Tigray, Ethiopia, 2016–2019 (N = 30, 300).**

| Variables | MTB Pos. N (%) | MTB Neg. N (%) | COR (95% CI) | P value | AOR (95%CI) | P value |
|---|---|---|---|---|---|---|
| **Gender** | | | | | | |
| Male | 1457(61) | 16014(57.4) | Ref | | Ref | |
| Female | 930 (39) | 11899(42.6) | 0.86(0.79–0.94) | <0.001* | 0.86(0.79–0.94) | <0.001* |
| **Age** | | | | | | |
| 18–29 | 940(39.4) | 6513(23.3) | Ref | | Ref | |
| 30–39 | 580(24.3) | 5720(20.5) | 0.7(0.63–0.78) | <0.001* | 0.7(0.63–0.78) | <0.001* |
| 40–49 | 388(16.3) | 4961(17.8) | 0.54(0.48–0.61) | <0.001* | 0.54(0.47–0.61) | <0.001* |
| 50–59 | 239(10) | 4098(14.7) | 0.4(0.35–0.47) | <0.001* | 0.4(0.35–0.47) | <0.001* |
| 60–69 | 144(6) | 3605(12.9) | 0.28(0.23–0.33) | <0.001* | 0.28(0.18–0.27) | <0.001* |
| 70–112 | 96(4) | 3016(10.8) | 0.22(0.22–0.18) | <0.001* | 0.22(0.18–0.27) | <0.001* |
| **HIV status (n = 665)** | | | | | | |
| Positive | 288(12.1) | 2387(89.2) | 1. 56 (1.37–1.78) | <0.001* | 1.54(1.33–1.72) | <0.001* |
| Negative | 377(15.8) | 3283(11.8) | Ref | | Ref | |
| **TB Treatment History** | | | | | | |
| New cases | 2195(92) | 26801(96) | Ref | | Ref | |
| Relapse | 168(7) | 1054(3.8) | 1.95(1.6–2.3) | <0.001* | 2.0(1.69–2.36) | <0.001* |
| Lost | 7(0.3) | 18(0.1) | 4.75(1.98–11.4) | <0.001* | 5.21(2.1–12.8) | <0.001* |
| Failure | 17(0.7) | 40(0.1) | 5.2(2.94–9.17) | <0.001* | 5.4(3.0–9.7) | <0.001* |

*: Statistically significant.

**Table 3. Frequency of RR-MTB among adult TB patients by sex, age, treatment history and HIV status in Eastern, Mekelle and Southern Zones of Tigray, Ethiopia, 2016–2019 (N = 2,387).**

| Variables | RR-MTB N (%) | Not RR-MTB N (%) | COR (95% CI) | P value | AOR (95%CI) | P value |
|---|---|---|---|---|---|---|
| **Gender** | | | | | | |
| Male | 125(58.1) | 1332(61.3) | Ref | | Ref | |
| Female | 90(41.9) | 840(38.7) | 1.1(0.86–1.52) | 0.36 | 1.1(0.82–1.46) | 0.54 |
| **Age** | | | | | | |
| 18–29 | 90(41.9) | 850(39.1) | Ref | | Ref | |
| 30–39 | 58(27) | 522(24) | 1.1(0.74–1.5) | 0.79 | 0.51(0.31–0.84) | 0.008* |
| 40–49 | 21(9.8) | 367(16.9) | 0.54(0.33–1.82) | 0.14 | 0.51 (0.45–1.30) | 0.19 |
| 50–59 | 32(14.9) | 207(9.5) | 1.5(0.95–2.25) | 0.085 | 0.64(0.32–1.3) | 0.22 |
| 60–69 | 9(4.2) | 135(6.2) | 0.6(0.31–1.3) | 0.20 | 0.47(0.19–1.2) | 0.12 |
| 70–112 | 5(2.5) | 91(4.2) | 0.52(0.21–1.3) | 0.17 | 0.65(0.33–1.34) | 0.29 |
| **HIV status (n = 55)** | | | | | | |
| Positive | 17(7.9) | 271(12.5) | 0.62(0.37–1.03) | 0.06 | 0.53(0.31–0.9) | 0.064 |
| Negative | 38(17.7) | 339(15.6) | Ref | | Ref | |
| **TB Treatment History** | | | | | | |
| New case | 176(81.9) | 2019(93) | Ref | | Ref | |
| Relapse | 34(15.8) | 134(6.2) | 2.9(1.94–4.37) | <0.001* | 3.26(2.14–4.97) | <0.001* |
| Lost | 1(0.5) | 6(0.3) | 1.9(.23-15-97) | 0.55 | 1.7(0.19–14.52) | 0.63 |
| Failure | 4(1.9) | 13(0.6) | 3.5(1.2–10.94) | 0.029 | 3.75(1.18–11.9) | 0.025* |

*: Statistically significant.

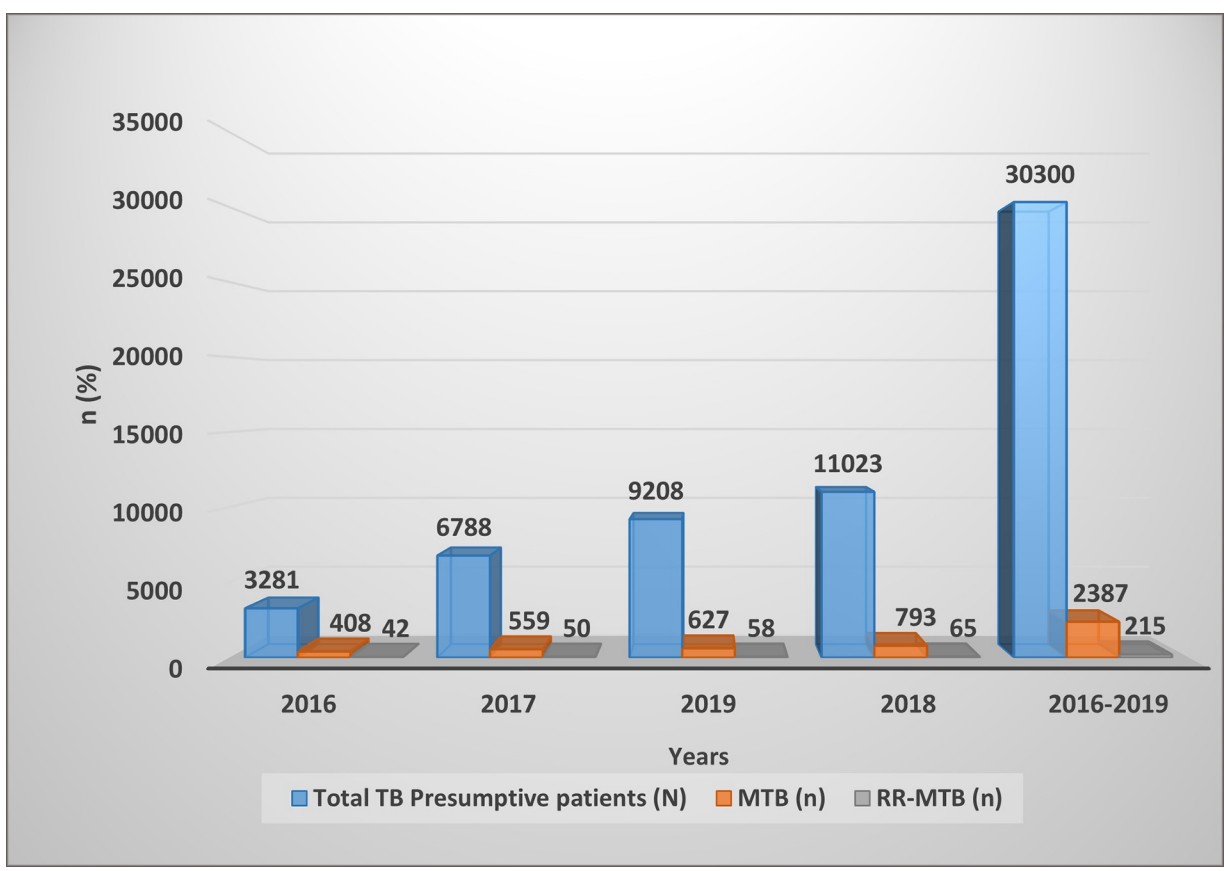

**Fig 2. Frequency of MTB and RR- MTB by study years.**

## Discussion

Availability of local epidemiological data on MTB and RR-MTB and identification of potentially predisposing factors is of paramount to design appropriate intervention strategies to control MTB. Overall, the frequency of MTB and RR-MTB in this study were 7.9% and 9%, respectively. MTB frequency (7.9%) in this study was more or less comparable with previous reports from Addis Ababa, [17] and Amhara region, [18]. However, our result was lower than studies conducted in Addis Ababa [11, 19], Southern Ethiopia [12], Somali region [20], Tigray region [15] and Oromia region [21], Amhara region [4, 22, 23], Congo [24], South Africa [25], Togo [26], Nigeria [27, 28], Korea [29], Pakistan [30], India [27, 31–34] and China [35]. However, our frequency was higher than the study conducted in Oromia region [36].

Possible reasons for the variations could be due to differences in methodological techniques (culture vs Xpert), study participants, study period, sample size, geography and TB control and prevention practices. For example, the high TB frequency in South Ethiopia [12], Oromia region [21], Amhara region [23], Tigray region [15], Congo [24], India [33] and Togo [26] was because their study participants were MDR presumptive patients (relapse, defaulter, lost and failure) unlike this study which included TB presumptive patients. Another possible reason for the high MTB recovery in the other studies such as Oromia region [36], Amhara region, [22], and Nigeria [28], used small sample size (small sampling could artificially generate higher prevalence rate).

The high prevalence of MTB in the studies from Somalia region [20], Pakistan [30], Bangladesh [37], India [32] and Togo [26] compared to our results reflects the higher disease burden in these countries. It could be due to the fact that their early reporting period, which was 2011 to 2014 where Xpert was initially indicated for patients with TB/HIV co-infection, presumptive MDR-TB and paediatrics TB patients. Whereas, this study was carried out from 2016 to 2019 where the method was used to all presumptive MTB patients.

The age range of our study participants was 18–112 years. Of these, participants whose age was 29 years or greater were less infected by MTB compared to the 18–29 years age groups (p< 0.05). Though there is no clear cut value for age, other studies reported higher prevalence in different age groups: 16–30 years [12], 25–34 years [22], while others reported no association between age and TB infection [2, 11, 21, 27]. Females were 14% times less likely [AOR = 0.86; 95% CI = 0.79, 0.94, p< 0.001] to have TB compared to males which was supported by other studies [4], Philippines [38] and North Sudan [36]. This could be probably due to males usually spending less time at home and have more frequent contacts with TB patients while females usually stay at home. Hence this could put males at more exposure to the disease [39].

On the other hand, previously treated patients (failure, relapse and lost to follow up) were more infected by MTB compared to new cases which was in line with a report by Adane *et al* [22]. The high TB prevalence in the previously treated highlights the need to give due attention in the DOT program as this may indicate high TB transmission to new TB cases in the community and in the case of relapse, the lack of TB treatment monitoring and control.

The distribution of RR-MTB is a big health problem in the study population. The frequency of RR-MTB (9%) in our study is in line with previous reports from Addis Ababa [11, 17], Amhara region, [4], Nigeria [27, 40], Korea [30], and India [32–34]. However, our frequency is lower than previous similar studies conducted in other parts of Ethiopia: Oromia region [21], Amhara region [23], Tigray region [14, 15], Congo [24], Nigeria [28], Togo [26], Russia [41], India [34], Bangladesh [37], Pakistan [31] and China [35]. Others have reported lower RR- TB prevalence in Southern Ethiopia [12], Amhara region [22], and Addis Ababa [19].

Possible reasons for the differences in the RR-MTB results could be due to variations in geography, methodology (sample size, method of diagnosis, study participants), study setting, study period and TB control practice. The high RR-MTB prevalence in Oromia region [21], Amhara region [23], Tigray region [15], Congo [24], India [33] and Togo [26] is due to the fact their study participants were suspected of MDR-TB (relapse, defaulter, lost and failure) unlike the presumptive TB patients which are included in this study. Low RR-MTB in study by Hamusse et al [36], could be due to the fact that the study was a community based study, not health service based one. The high RR-MTB reports from Somali region [20], Pakistan [31], Bangladesh [37], India [27] and Togo [26] compared to this report might be due to the difference in the study period and the scope to use Xpert for TB diagnosis. These studies were conducted between 2011 and 2014, where Xpert assay was indicated only for patients with presumptive MDR-TB. This study however, included data from 2016 to 2019 where Xpert-MTB/RIF Assay was recommended for all TB suspected patients.

In the present study, gender was not associated with RR-MTB; however, others [11, 23, 41] have indicated more RR-MTB among females compared to males. In contrary to this, other studies have reported more RR-MTB infection among males [14, 27, 40]. Similarly, age groups of 30–39 years in this study were less infected by RR-MTB compared to the other age group which was similar to other studies [23, 40]. The other associated factor with RR-MTB in this study was previous TB treatment. This is in agreement with previous reports [11, 21, 23, 27, 41]. The strong association of rifampicin resistance MTB with previous treatment highlights

the need for coordinated work of stake holders so as to improve the monitoring of treatment to reduce the emergence of circulating drug resistant MTB strains in the community.

This study has also tried to see the trend of MTB and RR-MTB through time. Accordingly, a significant decrease in the percentage of MTB frequency while the actual number of MTB detection increases show that the regional government and stakeholders have to perform well to tackle tuberculosis in the region. Though it was not statistically significant, the absolute number of RR-MTB positive cases shows an increment from 2016 to 2018.

This multicenter health facility based study was held in Tigray regional state and collected a large sample size. This is believed to give an updated information on the frequency of TB and RR-MTB to the regional and national governments. However, this study has limitations. First, the study was carried out only in one region of the nation (Tigray); the economic and regional disparities limited the generalization of the result. Second, we couldn't do microbiological confirmation of the Xpert positive MTB and RR-MTB, phenotypic rifampicin resistance and resistance to other anti-TB drugs because of the retrospective nature of the study. Third, we could not get information on contact history of MDR -TB and TB, education, and living conditions of patients; thus, we were unable to show the associations between these factors with our outcome variables.

## Conclusion

The frequency of MTB and RR-MTB in this study were 7.9% and 9%, respectively. Age (18–29 years), HIV positive and previous TB treatment history were significantly associated with high MTB, whereas gender (being female) was associated with low MTB. While rifampicin resistant-MTB was more prevalent among relapse and failure cases; it was lower among the age group of 30–39 years. The strong association MTB and RR-MTB with previous treatment highlights the need for more attention in TB treatment and monitoring program in the study area. Though frequency of MTB shows a decreasing trend over the study period, the prevalence still shows that more works should be done to further combat MTB associated morbidities and mortalities in the study area.

## Acknowledgments

We would like to thank all the hospital directors and laboratory staff of the study hospitals for their co-operation in allowing the researchers to access the records and extract the data.

## Author Contributions

**Conceptualization:** Araya Gebreyesus Wasihun, Tsehaye Asmelash Dejene, Genet Gebrehiwet Hailu.

**Formal analysis:** Araya Gebreyesus Wasihun, Genet Gebrehiwet Hailu.

**Methodology:** Araya Gebreyesus Wasihun.

**Visualization:** Tsehaye Asmelash Dejene.

**Writing – original draft:** Araya Gebreyesus Wasihun, Tsehaye Asmelash Dejene.

**Writing – review & editing:** Araya Gebreyesus Wasihun, Tsehaye Asmelash Dejene, Genet Gebrehiwet Hailu.

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
