## [Decision Letter · Decision Letter 0]

8 Jun 2020

PONE-D-20-09432

MTB and Rifampicin Resistance TB using Gene-Xpert-MTB/RIF Assay among Adult Presumptive Tuberculosis Patients in Tigray, Northern Ethiopia: a cross sectional study

PLOS ONE

Dear Dr. Wasihun,

Thank you for submitting your manuscript to PLOS ONE. After careful consideration, we feel that it has merit but does not fully meet PLOS ONE’s publication criteria as it currently stands. Therefore, we invite you to submit a revised version of the manuscript that addresses the points raised during the review process.

We look forward to receiving your revised manuscript.

Kind regards,

Shampa Anupurba, MD

Academic Editor

PLOS ONE

Journal Requirements:

Additional Editor Comments (if provided):

Reviewers' comments:

Reviewer's Responses to Questions

**Comments to the Author**

1. Is the manuscript technically sound, and do the data support the conclusions?

Reviewer #1: Partly

Reviewer #2: Yes

Reviewer #3: Partly

2. Has the statistical analysis been performed appropriately and rigorously? 

Reviewer #1: No

Reviewer #2: Yes

Reviewer #3: Yes

3. Have the authors made all data underlying the findings in their manuscript fully available?

Reviewer #1: Yes

Reviewer #2: Yes

Reviewer #3: Yes

4. Is the manuscript presented in an intelligible fashion and written in standard English?

Reviewer #1: No

Reviewer #2: No

Reviewer #3: Yes

5. Review Comments to the Author

Reviewer #1: Despite you try TO deal with the current national and global issue, the drug resistance TB, but your manuscript is not well written. I presented some of my comments, but many still left,

General comments

- Your writing is not consistent terms. E,g since xpert detect only mycobacterium tuberculosis it should be written as “RR_MTB,” but you randomly wrote as RR-TB, RR-MTB. It should be consistent though out the manuscript

- There may vocabulary / grammar errors. Incases MTP instead MTB

- You use your non standard terminologies: RR-TB positive AND RR-TB negative (table 1). RR_MTB DETECTED OR RR_MTB NOT DETECTED

- The texts/ paragraph with our references e.g “….Most studies in Ethiopia used the conventional culture and sensitivity methods not the automated Xpert MTB/RIF assay…” (Introduction). There are also reliable references. E.g there reference for population size of 6,960,000 of the region is the study from south Africa “Group Accuracy of the Xpert MTB / RIF test for the diagnosis of pulmonary tuberculosis in children admitted to hospital in Cape Town , South Africa : a descriptive study” . All the reference should be checked and reliable

Abstract

1. Part of the conclusion is not drawn from your data. In cases you try to highlights the need of a coordinated work in health education despite your data not support it

Methodology

1. The study setting is confusing : your describe that there 12 general hospitals, 22 primary hospitals,…… in the region” . You collect data from the five general hospitals and one compressive hospital. On the parag 3 you’re said “There are two general hospitals and one primary hospital which are not included in the study”.

2. Inclusion criteria;-despite you used secondary data (log book) but you exclude children and sample other than the sputum. Why ?

3. Laboratory Processing: it says “Samples were collected before the patients started anti-TB treatment” but on your result there are patents with failure and lost who already on treatment but sample collected for the indication probably as per national protocol

4. HIV testing: you describe the algorithm which could potential be modified as per the change in technology without reference.

5. Quality control: “… the researchers checked and confirmed that the Gene Xpert MTB/RIF assay was done using standard operating procedures”. How it possible for the secondary data? You can obtain SOP but how you know whether they followed or not. Such unnecessary should be modified

Result

1. Sociology-demographic, Clinical characteristics and TB results

- Using median instead of the mean is more appropriate for your study as the age range 18-112, which seems having extreme age like 112.

- The writing style not consistent. E.g (17,471; 57.7 %), 28,996 (95.7 %), 7.9 %

2. Associated Risk Factors of MTB or RR_ MTB Infections

- Interpretation of the statistical finding wrong. E.G for AOR of 0.86, your describe as “…females were 86 % times less likely to be infected by TB compared to males”. Rather is should be … female has 26% less likely infected by TB compared to males”. Many other examples found in your result section while you interpreted the associated Risk Factors of for MTB AND RR –TB. This misleading interpretation also reflected in your discussion part.

Reviewer #2: Abstract

Methods

- The authors may delete the period when the data extraction was done (Oct - Dec 2019) given that have also included the focus period Jan 2016 - Dec 2019.

Results

- The authors mention "high TB infection" which is confusing and yet the study focused on MTB (TB) and RR TB (MDR TB).

- The interpretation of the first part of the statement "Likewise rifampicin resistant was more prevalent .....(p<0.05) should be aligned to table 3 findings.

Conclusion

-The authors need to include a statement on whether the prevalence was high or low - this may eliminate the repetition of figures which are already listed under the results section.

- The final recommendation listed is not clear

Main text

Introduction

- Reference 1 - There is a more recent global TB report 2019

- Paragraph 2 - There are 30 MDR TB high burden countries - The authors should replace reference 9 with a more recent reference

- The text listed in paragraph 2 is not clear and requires revision

Study setting

- The authors include detailed geographical information such as longitude and latitude which may not be relevant and can be deleted.

- The authors may delete the period when the data extraction was done (Oct - Dec 2019) given that have also included the focus period Jan 2016 - Dec 2019.

- The text on the selected and non selected sites can be shortened. This will make it easier to understand.

Inclusion criteria

- Include "presumptive TB" before adult patients - this is the study population

Outcome variables

- Delete prevalence. Outcome variable is TB and RR TB

Data collection

- It is not clear whether the patient records were paper based or electronic.

Ethical consideration

- This text can be shortened

Results

Table 1

- The total for the variables under TB treatment history is (41,300) is more than the sample size (30,300). The figures need to be verified

Associated risk factors for MTB infections

- The authors use "infected by TB" which may confuse the reader. They may consider using "TB" e.g. ...."less likely to have TB" as opposed to "less likely to be infected by TB"

- The authors do not need to list all the odds for the different age groups since they are reflected in table 2. The odds show a decreasing trend and can be summarized as such.

- The authors do not need to list all the odds for the "TB treatment history" groups since they are reflected in table 2. This may be summarized by mentioning that the odds of TB were higher among previously treated patients.

Table 2

- Column proportions would be more informative especially when comparing proportions across the independent variable groups by the outcome variable.

Table 3

- Column proportions would be more informative especially when comparing proportions across the independent variable groups by the outcome variable.

- Does changing "HIV negative" to the ref group in the bi-variate and multi-variate analysis change the results. It would be good to run the analysis for comparison.

Prevalence of MTB and RR-TB by years

- Figure 2 - Does "total patients refer to "presumptive TB"? This should be clarified

- While the authors state that the prevalence of TB has reduced over time, it is important to also make a comment on the absolute numbers which have increased over time. The numbers for RR TB are not included on the figure.

Discussion

- Paragraph 2 - The authors list that their findings are more or less comparable and then include contradicting statements thereafter. They list all the proportions for the various studies in reference which might not be necessary. The authors may include a summary statement and simply quote the references.

Paragraph 2 - Text can be reduced. Authors list all proportions from other studies which makes the text really long. The authors may include a summary statement and simply quote the references.

- Paragraph 2 - It is not clear why the authors compare their findings to a study conducted in Uganda (29) which focused on children. They excluded children. Furthermore, the two populations differ in disease patterns which would also . It makes more sense for the authors to limit the comparison to adult studies.

- Paragraph 3 - The high prevalence of TB in studies from Somalia, Pakistan, Bangladesh..... reflects the higher disease burden in these countries.

- Paragraph 3 - See comments above on "TB infection" and "more infected by TB"

Conclusion

- See comments above (abstract)

Reviewer #3: Comments to the Author

Manuscript Number: PONE-D-20-09432

Title: MTB and Rifampicin Resistance TB using Gene-Xpert-MTB/RIF Assay among Adult Presumptive Tuberculosis Patients in Tigray, Northern Ethiopia: a cross sectional study

This is an interesting study that potentially represents the prevalence of tuberculosis and multidrug resistant tuberculosis in in Tigray, Northern Ethiopia. This study also giving information about increasing trend of multiple drug resistance against TB which an alrming condition.

General comments

This study is well described, however there are certain limitations in the study that need to be addressed.

Title: Title is not matched with study. Title could be better like “ Frequency of MTB and Rifampicin Resistance TB using Xpert-MTB/RIF Assay among Adult Presumptive Tuberculosis Patients in Tigray, Northern Ethiopia: a cross sectional study” instead of “MTB and Rifampicin Resistance TB using Gene-Xpert-MTB/RIF Assay among Adult Presumptive Tuberculosis Patients in Tigray, Northern Ethiopia: a cross sectional study”

Abstract:

This is not Prevalence study so replace prevalence words by frequency

Replace Gene-Xpert-MTB/RIF Assay by Xpert-MTB/RIF Assay

Methods:

Please correct timing of study because you wrote October 2019 to December 2019 in one line and January 2016 to December 2019 in other line.

Results:

Line number 6, Please write number out of total and then write percentage in bracket. For example you wrote in line number 3, 17,471 (57.7 %) were males.

It would be better if you shows the significant value with males and previous history that how much it is significant

Conclusion: Don’t start paragraph with number like 7.9%

Introduction:

It would be better if you define 1st susceptible and resistant tuberculosis, Rifampecin resistant and then MDR-TB.

You can help from this article (Javaid A, Ullah I, Masud H, Basit A, Ahmad W, Butt ZA, Qasim M. Predictors of poor treatment outcomes in multidrug-resistant tuberculosis patients: a retrospective cohort study. Clinical Microbiology and Infection. 2018 Jun 1;24(6):612-7).

Paragraph 3, line 6. Write RR-TB in full instead of abbreviation 1st and check thought out the manuscript.

Materials and Methods

Please make a table or box and write all the definition like Variables, outcomes relapse, failure, relapse etc

Results:

Please go through overall papers as some paragraphs are confusing and not clear. Rephrase it like “According to the results of this study, the overall, prevalence of TB and RR- TB were 7.9 % and 9 %, respectively” and “As can be seen in Table 2, females were 86 % times less likely [Adjusted Odds Ratio (AOR) =0.86; 95 % CI= 0.79, 0.94, p= 0.000] to be infected by TB compared to males.

Discussion:

Discussion is overall good but need to be slightly modify it by grammatically

6. PLOS authors have the option to publish the peer review history of their article (what does this mean?). If published, this will include your full peer review and any attached files.

Reviewer #1: No

Reviewer #2: No

Reviewer #3: Yes: Dr Irfan Ullah

---

## [Author Response · Author response to Decision Letter 0]

28 Jul 2020

Point-By-Point Response Letter to Reviewers Comments

Dear Editor,

Greetings,

First and foremost, we thank you for giving us the chance to revise our manuscript. We really appreciate the interest of the reviewers to our paper, and their critical and important comments which made us learn a lot. We have revised the manuscript and reviewers’ comments and questions are addressed in highlighted in the revised manuscript.

Kind regards,

Araya GebreyesusWsihun, on behalf of the research members. 

Reviewer #1: 

Dear Reviewer,

I, on behalf of the team would like to thank you for reviewing our manuscript critically and in a detail way. Your constructive comments not only help us enrich the manuscript, but also made us learn a lot on how one should review articles. We have tried to address your comments as much as we could. Thank you once again for making us learn. 

Reviewer’s general comments 

Despite you try TO deal with the current national and global issue, the drug resistance TB, but your manuscript is not well written. I presented some of my comments, but many still left,- Your writing is not consistent terms. E,g since xpert detect only mycobacterium tuberculosis it should be written as “RR_MTB,” but you randomly wrote as RR-TB, RR-MTB. It should be consistent though out the manuscript

Author’s response: Corrected.

- Reviewer’s comments: There may vocabulary / grammar errors. Incases MTP instead MTB

Author’s response: Corrected. 

- Reviewer’s comments: You use your nonstandard terminologies: RR-TB positive AND RR-TB negative (table 1). RR_MTB DETECTED OR RR_MTB NOT DETECTED

Author’s response: Corrected.

Reviewer’s comments: The texts/ paragraph with our references e.g “….Most studies in Ethiopia used the conventional culture and sensitivity methods not the automated Xpert MTB/RIF assay…” (Introduction). There are also reliable references. E.g there reference for population size of 6,960,000 of the region is the study from south Africa “Group Accuracy of the Xpert MTB / RIF test for the diagnosis of pulmonary tuberculosis in children admitted to hospital in Cape Town , South Africa : a descriptive study” . All the reference should be checked and reliable

Author’s response: Thank you very much for your comments and reference you provided. However, our statement is specific to Ethiopia; not referring to Africa or broader studies. As can be seen, in the discussion part, we have compared our result with similar studies on Xpert for other parts. 

Abstract

Reviewer’s comments: Part of the conclusion is not drawn from your data. In cases you try to highlights the need of a coordinated work in health education despite your data not support it

Author’s response: Corrected.

Methodology

 Reviewer’s comments: The study setting is confusing: your describe that there 12 general hospitals, 22 primary hospitals, in the region”. You collect data from the five general hospitals and one compressive hospital. On the page 3 you’re said “There are two general hospitals and one primary hospital which are not included in the study”.

Author’s response: Corrected 

Reviewer’s comments: 2. Inclusion criteria;-despite you used secondary data (log book) but you exclude children and sample other than the sputum. Why?

Author’s response: Thank you very much for raising very important question. Given our objective was to determine prevalence of pulmonary TB among adults, where TB is more prevalent, we did not include children and other samples. But as you pretty mentioned, we believe their inclusion could have given some additional information on RR-MTB and MTB in the study area.

Reviewer’s comments: 3. Laboratory Processing: it says “Samples were collected before the patients started anti-TB treatment” but on your result there are patents with failure and lost who already on treatment but sample collected for the indication probably as per national protocol

Author’s response: Thank you very much for such constructive comments indeed. We have removed such controverting sentence and corrected accordingly.

Reviewer’s comments: 4. HIV testing: you describe the algorithm which could potential be modified as per the change in technology without reference.

Author’s response: Thank you for your important comment. We have now corrected and rewritten as: Rapid HIV test was done according to the national algorithm of the Federal Ministry of Health of Ethiopia.

Reviewer’s comments: 5. Quality control: “… the researchers checked and confirmed that the Gene Xpert MTB/RIF assay was done using standard operating procedures”. How it possible for the secondary data? You can obtain SOP but how you know whether they followed or not. Such unnecessary should be modified

Author’s response: As you clearly mentioned, it is possible to get the SOP and we indeed obtained it, but it is not possible to confirm whether or not followed. Hence we corrected such ambiguity sentences in the revised version as per the given comments. 

Result

Reviewer’s comments: 1. Sociology-demographic, Clinical characteristics and TB results 

Using median instead of the mean is more appropriate for your study as the age range 18-112, which seems having extreme age like 112.

Author’s response: Though there are age outliers, age distribution in the histogram shows normal distribution, hence we used the mean. But as per your comment, we changed to median which was 40.65 (interquartile 29.4 -56.5 years).

Reviewer’s comments: The writing style not consistent. E.g (17,471; 57.7 %), 28,996 (95.7 %), 7.9 %

Author’s response: Corrected as N (%).

 2. Associated Risk Factors of MTB or RR_ MTB Infections

Reviewer’s comments: - Interpretation of the statistical finding wrong. E.G for AOR of 0.86, your describe as “…females were 86 % times less likely to be infected by TB compared to males”. Rather is should be … female has 26% less likely infected by TB compared to males”. Many other examples found in your result section while you interpreted the associated Risk Factors of for MTB AND RR –TB. This misleading interpretation also reflected in your discussion part.

Author’s response: Thank you very much. We have corrected the error in interpretation. 

Reviewer #2: Abstract

Dear Reviewer,

I, on behalf of the team would like to thank you for reviewing our manuscript critically and in a detail way. Your constructive comments not only help us enrich the manuscript, but also made us learn a lot on how one should review articles. We have tried to address your comments as much as we could. Thank you once again for making us learn a lot. 

Methods

Reviewer’s comments: - The authors may delete the period when the data extraction was done (Oct - Dec 2019) given that have also included the focus period Jan 2016 - Dec 2019.

Author’s response: Thank you for your critical comments, we have deleted. 

Results

Reviewer’s comments: - The authors mention "high TB infection" which is confusing and yet the study focused on MTB (TB) and RR TB (MDR TB).

Author’s response: Corrected. 

Reviewer’s comments: - The interpretation of the first part of the statement "Likewise rifampicin resistant was more prevalent .....(p<0.05) should be aligned to table 3 findings.

Author’s response: Corrected as per the recommendation. 

Conclusion

Reviewer’s comments: -The authors need to include a statement on whether the prevalence was high or low - this may eliminate the repetition of figures which are already listed under the results section.

Author’s response: We have corrected. 

Reviewer’s comments: - The final recommendation listed is not clear

Author’s response: Made clear in the revised manuscript

Main text

Introduction

Reviewer’s comments: - Reference 1 - There is a more recent global TB report 2019

Author’s response: We have use the 2019 reference. 

Reviewer’s comments: - Paragraph 2 - There are 30 MDR TB high burden countries - The authors should replace reference 9 with a more recent reference

Author’s response: Corrected. 

Reviewer’s comments: - The text listed in paragraph 2 is not clear and requires revision

Author’s response: Corrected 

Study setting

Reviewer’s comments: - The authors include detailed geographical information such as longitude and latitude which may not be relevant and can be deleted.

Author’s response: Deleted. 

Reviewer’s comments: - The authors may delete the period when the data extraction was done (Oct - Dec 2019) given that have also included the focus period Jan 2016 - Dec 2019.

Author’s response: Deleted

Reviewer’s comments: - The text on the selected and non-selected sites can be shortened. This will make it easier to understand.

Author’s response: Shortened

Inclusion criteria

Reviewer’s comments: - Include "presumptive TB" before adult patients - this is the study population

Author’s response: Included

Outcome variables

Reviewer’s comments: - Delete prevalence. Outcome variable is TB and RR TB

Author’s response: Deleted. 

Data collection

Reviewer’s comments: - It is not clear whether the patient records were paper based or electronic.

Author’s response: Corrected as paper based (registration books) 

Ethical consideration

Reviewer’s comments: - This text can be shortened

Author’s response: Corrected. 

Results

Table 1

Reviewer’s comments: - The total for the variables under TB treatment history is (41,300) is more than the sample size (30,300). The figures need to be verified

Author’s response: Thank you for your critical reviewing our paper. There was type error where ‘ 2’ was added. Now corrected. 

Associated risk factors for MTB infections

Reviewer’s comments: - The authors use "infected by TB" which may confuse the reader. They may consider using "TB" e.g. ...."less likely to have TB" as opposed to "less likely to be infected by TB"

Author’s response: Corrected as per the comments throughout the manuscript. 

Reviewer’s comments: - The authors do not need to list all the odds for the different age groups since they are reflected in table 2. The odds show a decreasing trend and can be summarized as such.

Author’s response: We have corrected and put in summary. 

Reviewer’s comments: - The authors do not need to list all the odds for the "TB treatment history" groups since they are reflected in table 2. This may be summarized by mentioning that the odds of TB were higher among previously treated patients.

Author’s response: Summarized.

Table 2

Reviewer’s comments: - Column proportions would be more informative especially when comparing proportions across the independent variable groups by the outcome variable.

Author’s response: Changed to column proportions 

Table 3

Reviewer’s comments: - Column proportions would be more informative especially when comparing proportions across the independent variable groups by the outcome variable.

Author’s response: Changed to column proportions

Reviewer’s comments: Does changing "HIV negative" to the ref group in the bi-variate and multi-variate analysis change the results. It would be good to run the analysis for comparison.

Author’s response: Changed and reanalyzed as per the given comments

Prevalence of MTB and RR-TB by years 

Reviewer’s comments: - Figure 2 - Does "total patients refer to "presumptive TB"? This should be clarified

Author’s response: Yes, and we have corrected in the figure. 

Reviewer’s comments: While the authors state that the prevalence of TB has reduced over time, it is important to also make a comment on the absolute numbers which have increased over time. The numbers for RR TB are not included on the figure.

Author’s response: We have corrected, and figure are put as number. 

Discussion

Reviewer’s comments: - Paragraph 2 - The authors list that their findings are more or less comparable and then include contradicting statements thereafter. They list all the proportions for the various studies in reference which might not be necessary. The authors may include a summary statement and simply quote the references.

Author’s response: Corrected. 

Reviewer’s comments: Paragraph 2 - Text can be reduced. Authors list all proportions from other studies which makes the text really long. The authors may include a summary statement and simply quote the references.

Author’s response: Shortened by deleting the figures. 

Reviewer’s comments: - Paragraph 2 - It is not clear why the authors compare their findings to a study conducted in Uganda (29) which focused on children. They excluded children. Furthermore, the two populations differ in disease patterns which would also. It makes more sense for the authors to limit the comparison to adult studies.

Author’s response: We have deleted the comparison made with children from Uganda 

Reviewer’s comments: - Paragraph 3 - The high prevalence of TB in studies from Somalia, Pakistan, Bangladesh..... reflects the higher disease burden in these countries.

Author’s response: Corrected. 

Reviewer’s comments: - Paragraph 3 - See comments above on "TB infection" and "more infected by TB"

Author’s response: Corrected.

Conclusion

Reviewer’s comments: - See comments above (abstract)

Author’s response: Corrected. 

Reviewer #3: 

Dear Reviewer,

I, on behalf of the team would like to thank you for reviewing our manuscript critically and in a detail way. Your constructive comments not only help us enrich the manuscript, but also made us learn a lot on how one should review articles. We have tried to address your comments as much as we could. Thank you once again for making us learn a lot. 

Manuscript Number: PONE-D-20-09432

Title: MTB and Rifampicin Resistance TB using Gene-Xpert-MTB/RIF Assay among Adult Presumptive Tuberculosis Patients in Tigray, Northern Ethiopia: a cross sectional study

This is an interesting study that potentially represents the prevalence of tuberculosis and multidrug resistant tuberculosis in in Tigray, Northern Ethiopia. This study also giving information about increasing trend of multiple drug resistance against TB which an alrming condition.

General comments

This study is well described, however there are certain limitations in the study that need to be addressed.

Reviewer’s comments: Title: Title is not matched with study. Title could be better like “ Frequency of MTB and Rifampicin Resistance TB using Xpert-MTB/RIF Assay among Adult Presumptive Tuberculosis Patients in Tigray, Northern Ethiopia: a cross sectional study” instead of “MTB and Rifampicin Resistance TB using Gene-Xpert-MTB/RIF Assay among Adult Presumptive Tuberculosis Patients in Tigray, Northern Ethiopia: a cross sectional study”

Author’s response: Thank you for your constructive comments. We have changed as per the comments. 

Abstract:

Reviewer’s comments: � This is not Prevalence study so replace prevalence words by frequency

Author’s response: Replaced.

Reviewer’s comments: � Replace Gene-Xpert-MTB/RIF Assay by Xpert-MTB/RIF Assay

Author’s response: Replaced throughout the paper

Methods:

Reviewer’s comments: � Please correct timing of study because you wrote October 2019 to December 2019 in one line and January 2016 to December 2019 in other line.

Author’s response: Corrected. 

Results:

Reviewer’s comments: � Line number 6, Please write number out of total and then write percentage in bracket. For example you wrote in line number 3, 17,471 (57.7 %) were males.

Author’s response: Corrected as N (%).

Reviewer’s comments: � It would be better if you shows the significant value with males and previous history that how much it is significant

Author’s response: Corrected. 

Reviewer’s comments: Conclusion: Don’t start paragraph with number like 7.9%

Author’s response: Corrected 

Introduction:

Reviewer’s comments: � It would be better if you define 1st susceptible and resistant tuberculosis, Rifampecin resistant and then MDR-TB.

You can help from this article (Javaid A, Ullah I, Masud H, Basit A, Ahmad W, Butt ZA, Qasim M. Predictors of poor treatment outcomes in multidrug-resistant tuberculosis patients: a retrospective cohort study. Clinical Microbiology and Infection. 2018 Jun 1;24(6):612-7).

Author’s response: we have defined MDR-TB in the introduction, we have also added Operational definition part in the methods. 

Reviewer’s comments: � Paragraph 3, line 6. Write RR-TB in full instead of abbreviation 1st and check thought out the manuscript.

Author’s response: Written.

Materials and Methods

Reviewer’s comments: � Please make a table or box and write all the definition like Variables, outcomes relapse, failure, relapse etc

Author’s response: Included in the operational definition.

Results:

Reviewer’s comments: Please go through overall papers as some paragraphs are confusing and not clear. Rephrase it like “According to the results of this study, the overall, prevalence of TB and RR- TB were 7.9 % and 9 %, respectively” and “As can be seen in Table 2, females were 86 % times less likely [Adjusted Odds Ratio (AOR) =0.86; 95 % CI= 0.79, 0.94, p= 0.000] to be infected by TB compared to males.

Discussion:

Author’s response: Corrected. 

Reviewer’s comments: Discussion is overall good but need to be slightly modify it by grammatically

Author’s response: Improved.

---

## [Decision Letter · Decision Letter 1]

20 Aug 2020

PONE-D-20-09432R1

Frequency of MTB and Rifampicin Resistance MTB using Xpert-MTB/RIF Assay among Adult Presumptive Tuberculosis Patients in Tigray, Northern Ethiopia: a cross sectional study

PLOS ONE

Dear Dr. Wasihun,

Thank you for submitting your manuscript to PLOS ONE. After careful consideration, we feel that it has merit but does not fully meet PLOS ONE’s publication criteria as it currently stands. Therefore, we invite you to submit a revised version of the manuscript that addresses the points raised during the review process.

In addition to the reviewer's comment, I noticed that you have missed out the following:

1. "You use your non standard terminologies: RR-TB positive AND RR-TB negative (table 1). RR_MTB DETECTED OR RR_MTB NOT DETECTED"-same changes should be made with respect to MTB DETECTED/NOT DETECTED

2. "Part of the conclusion is not drawn from your data. In cases you try to highlights the need of a coordinated work in health education despite your data not support it".This has been omitted from the abstract,but is still present in the conclusion of the manuscript.

We look forward to receiving your revised manuscript.

Kind regards,

Shampa Anupurba, MD

Academic Editor

PLOS ONE

Reviewers' comments:

Reviewer's Responses to Questions

**Comments to the Author**

1. If the authors have adequately addressed your comments raised in a previous round of review and you feel that this manuscript is now acceptable for publication, you may indicate that here to bypass the “Comments to the Author” section, enter your conflict of interest statement in the “Confidential to Editor” section, and submit your "Accept" recommendation.

Reviewer #1: All comments have been addressed

2. Is the manuscript technically sound, and do the data support the conclusions?

Reviewer #1: Yes

3. Has the statistical analysis been performed appropriately and rigorously? 

Reviewer #1: (No Response)

4. Have the authors made all data underlying the findings in their manuscript fully available?

Reviewer #1: Yes

5. Is the manuscript presented in an intelligible fashion and written in standard English?

Reviewer #1: No

6. Review Comments to the Author

Reviewer #1: Thank you for dressing most of my comments and suggestions in the first draft. I have seen good improvement in the manuscript. However, I have some mirror correction

Abstract

1. Change the mean to the median in the abstract, to make it consistent

2. Change “The overall frequency of TB was…” to “The overall frequency of MTB was…’’. Pleases use MTB as Xpert MTB/RIF detects only MTB and its RIF resistant strain but not other species of tuberculosis. Make it consistent in part too

Introduction

1. “Its drug-resistant strain called multidrug-resistant mycobacterium tuberculosis (MDR-MTB)….”. This sentence looks confusing as mycobacterium tuberculosis, not the only MDR strain. Please modify it

2. Last paragraph… ‘’ Besides, they were done using culture and drug susceptibility testing methods….’’.I think this is no limitation rather it is good as it detects other species/strain of TB. Good to modify or remove it

Methods

1. Outcome variable: not “TB and RR-MTB among presumptive TB patients” rather MTB and RR-MTB among presumptive TB patients

2. Operational definition

o MDR-TB: is not “Isolate of M. tuberculosis…..” Not only M. tuberculosis, why other species? modify it

o Rifampicin-resistant TB (RR-MTB) change to : Rifampicin-resistant TB (RR-TB)

Result

Associated Risk Factors of MTB Infections

1 “………1.46 times [AOR= 1.46; 95% CI =1.29, 1.57, p <0.001] times more likely to have………” correct as “………1.46 times [AOR= 1.46; 95% CI =1.29, 1.57, p <0.001] more likely to have………”

2 Change TB to MTB

3. Frequency of MTB and RR- MTB by study years

1. Change “Figure 2 compares the frequency of TB and RR-MTB by study years” to Figure 2 compares the frequency of MTB and RR-MTB by study years. Please do the same for others in this paragraph. Do mix TB and MTB

Discussion

1. The same research finding “Addis Ababa, [17]” in opposite comparation. ‘MTB frequency (7.9%) in this study was more or less comparable with previous reports from Addis Ababa, [17]……………………..”. “However, our frequency was higher than studies conducted in Addis Ababa [17]”. please modify the reference “17’’

2. . change ‘’For Example……” , For example,

3. The sentence (paragraph 4) is opposite to you finding “Of these, participants whose age was 29 years or greater were more infected by TB compared to the 18-29 years age groups (p< 0.05).” pleases modify it

4. Whereas, this study was carried out from 2016 to 2019 where the method was used to all presumptive TB patients. This sentence repeated pleases good to explain with other explanation

5. Correct 3039 to 30-39

6. “Accordingly, the number of TB suspected patients who visited the hospitals significantly increased from 3281 in 2016 to 11023 in 2018. Similarly, the absolute number of TB positive patients has also increased from 408 in 2016 to 793 in 2018. However, the actual percent of TB frequency showed a significant decrease from (12.4%) in 2016 to (6.8%) in 2019” this is the direct copy of the result, pleases do not repeat/ copy rather discuss the concept. for example, the concept can be summerazised like this “A significant decrease in the percentage of MTB frequency while the actual number of TB detection increases show that the regional government and stakeholders have to perform well to tackle tuberculosis in the region

7. PLOS authors have the option to publish the peer review history of their article (what does this mean?). If published, this will include your full peer review and any attached files.

---

## [Author Response · Author response to Decision Letter 1]

27 Aug 2020

Point-By-Point Response Letter to Reviewers Comments

Dear Editor,

Greetings,

First and foremost, we thank for giving a chance to revise our paper. We really appreciate the interest of the reviewers to our paper, and we also appreciate for their critical and important comments and recommendations. We have revised the manuscript and reviewer’s comments and questions are addressed in highlighted in the revised manuscript.

Kind regards,

Araya Gebreyesus Wsihun (PhD)

Editor’s comment 

In addition to the reviewer's comment, I noticed that you have missed out the following:

1. "You use your nonstandard terminologies: RR-TB positive AND RR-TB negative (table 1). RR_MTB DETECTED OR RR_MTB NOT DETECTED"-same changes should be made with respect to MTB DETECTED/NOT DETECTED

Response: Corrected 

2. "Part of the conclusion is not drawn from your data. In cases you try to highlights the need of a coordinated work in health education despite your data not support it". This has been omitted from the abstract, but is still present in the conclusion of the manuscript.

 Response: Now corrected 

Reviewer #1: Thank you for dressing most of my comments and suggestions in the first draft. I have seen good improvement in the manuscript. However, I have some mirror correction

Abstract

1. Change the mean to the median in the abstract, to make it consistent

Response: Changed

2. Change “The overall frequency of TB was…” to “The overall frequency of MTB was…’’. Pleases use MTB as Xpert MTB/RIF detects only MTB and its RIF resistant strain but not other species of tuberculosis. Make it consistent in part too

Response: Corrected

Introduction

1. “Its drug-resistant strain called multidrug-resistant mycobacterium tuberculosis (MDR-MTB)….”. This sentence looks confusing as mycobacterium tuberculosis, not the only MDR strain. Please modify it

Response: Corrected

2. Last paragraph… ‘’ Besides, they were done using culture and drug susceptibility testing methods….’’.I think this is no limitation rather it is good as it detects other species/strain of TB. Good to modify or remove it

Response: Removed

Methods

1. Outcome variable: not “TB and RR-MTB among presumptive TB patients” rather MTB and RR-MTB among presumptive TB patients

Response: Corrected

2. Operational definition

o MDR-TB: is not “Isolate of M. tuberculosis ” Not only M. tuberculosis, why other species? modify it

Response: Modified 

o Rifampicin-resistant TB (RR-MTB) change to : Rifampicin-resistant TB (RR-TB)

Response: Corrected

Result

Associated Risk Factors of MTB Infections

1 “………1.46 times [AOR= 1.46; 95% CI =1.29, 1.57, p <0.001] times more likely to have………” correct as “………1.46 times [AOR= 1.46; 95% CI =1.29, 1.57, p <0.001] more likely to have………”

Response: Corrected

2 Change TB to MTB

Response: Changed

3. Frequency of MTB and RR- MTB by study years

1. Change “Figure 2 compares the frequency of TB and RR-MTB by study years” to Figure 2 compares the frequency of MTB and RR-MTB by study years. Please do the same for others in this paragraph. Do mix TB and MTB

Response: Corrected

Discussion

1. The same research finding “Addis Ababa, [17]” in opposite cooperation. ‘MTB frequency (7.9%) in this study was more or less comparable with previous reports from Addis Ababa, [17]……………………..”. “However, our frequency was higher than studies conducted in Addis Ababa [17]”. please modify the reference “17’’

Response: Corrected

2. Change ‘’For Example……” , For example,

Response: Changed

3. The sentence (paragraph 4) is opposite to you finding “Of these, participants whose age was 29 years or greater were more infected by TB compared to the 18-29 years age groups (p< 0.05).” pleases modify it

Response: Modified 

4. Whereas, this study was carried out from 2016 to 2019 where the method was used to all presumptive TB patients. This sentence repeated pleases good to explain with other explanation

Response: Corrected

5. Correct 3039 to 30-39

Response: Corrected

6. “Accordingly, the number of TB suspected patients who visited the hospitals significantly increased from 3281 in 2016 to 11023 in 2018. Similarly, the absolute number of TB positive patients has also increased from 408 in 2016 to 793 in 2018. However, the actual percent of TB frequency showed a significant decrease from (12.4%) in 2016 to (6.8%) in 2019” this is the direct copy of the result, pleases do not repeat/ copy rather discuss the concept. for example, the concept can be summerazised like this “A significant decrease in the percentage of MTB frequency while the actual number of TB detection increases show that the regional government and stakeholders have to perform well to tackle tuberculosis in the region

Response: summarized as per the comment

---

## [Editor Report · Decision Letter 2]

10 Sep 2020

PONE-D-20-09432R2

Frequency of MTB and Rifampicin Resistance MTB using Xpert-MTB/RIF Assay among Adult Presumptive Tuberculosis Patients in Tigray, Northern Ethiopia: a cross sectional study

PLOS ONE

Dear Dr. Wasihun,

Thank you for submitting your manuscript to PLOS ONE. After careful consideration, we feel that it has merit but does not fully meet PLOS ONE’s publication criteria as it currently stands. Therefore, we invite you to submit a revised version of the manuscript that addresses the points raised during the review process.

We look forward to receiving your revised manuscript.

Kind regards,

Shampa Anupurba, MD

Academic Editor

PLOS ONE

Additional Editor Comments (if provided):

Mycobacterium tuberculosis should be mentioned as italics throughout the manuscript.

Abstract- Background 1st line-Delete 'Mycobacterium tuberculosis (MTB) and its'

Abstract- 4th line under Results- Please write median age 'of' 40.65 instead of 'was'

Abstract- Results-HIV unknown status cannot be associated with high MTB. This does not carry any meaning and should also be deleted from the entire manuscript.The statistical significance may not be shown in Table 1.

Methods- Study Setting- Delete 'the' in the sentence 'In this study, general hospitals which introduced Xpert since the 2016 were included.'

'intermediate' Xpert MTB/RIF results should be changed to 'indeterminate' throughout the manuscript.

Inclusion criteria- 'We include all presumptive TB'- change include to included

Table 1- Under MTB result, mention detected or not detected instead of positive /negative.

Discussion- 2nd para,6th line- 'unlike to this study'-delete 'to'

3rd para, 2nd line-'compared to our results which reflects the' delete 'which'

5th para,1st line-'On the other hand, previous treated patients'- write 'previously'

7th para, 2nd line-write geography instead of geographical

'This study; however, included data from 2016 and 2019'- replace 'and' by 'to', delete semi colon

'Xpert-MTB/RIF Assay is recommended to all TB suspected patients'- replace 'is' by 'was'. Also, delete 'with less likelihood of

having RR-MTB as the MDR-TB suspected patients used in the other studies'.

---

## [Author Response · Author response to Decision Letter 2]

12 Sep 2020

Point-by-point response letter to Editor’s comments

Dear Dr. Shampa Anupurba,

Warm greetings,

First and foremost, we thank for giving a chance to revise our paper. We really appreciate the interest of the editor to our paper, and we also appreciate for the critical and important comments and recommendations. We have revised the manuscript and editor’s comments in highlighted in the revised manuscript.

Kind regards,

Araya Gebreyesus Wasihun (PhD)

 Editor’s comment: Mycobacterium tuberculosis should be mentioned as italics throughout the manuscript.

Response: Done

Editor’s comment: Abstract- Background 1st line-Delete 'Mycobacterium tuberculosis (MTB) and its'

Response: Done 

Editor’s comment: Abstract- 4th line under Results- Please write median age 'of' 40.65 instead of 'was'

Response: Done 

Editor’s comment: Abstract- Results-HIV unknown status cannot be associated with high MTB. This does not carry any meaning and should also be deleted from the entire manuscript. The statistical significance may not be shown in Table 1.

Response: Dear Editor, we really thank you for the critical comment. Patients with unknown HIV status means that there was no any data on HIV result in the registration book for the patient. Which means the patient may be positive or negative. Again in table one HIV positivity was associated with high MTB. Though we cannot be sure, the HIV result of the patients might be positive. Now we deleted the row for Unknown and we made the regression analysis between the HIV positive and Negative in both tables as (n=655 for table 2) and (n=55 for table 3). 

Editor’s comment: Methods- Study Setting- Delete 'the' in the sentence 'In this study, general hospitals which introduced Xpert since the 2016 were included.'

Response: Deleted

Editor’s comment: 'Intermediate' Xpert MTB/RIF results should be changed to 'indeterminate' throughout the manuscript.

Response: Changed 

Editor’s comment: Inclusion criteria- 'We include all presumptive TB'- change include to included

Response: Changed 

Editor’s comment: Table 1- Under MTB result, mention detected or not detected instead of positive /negative.

Response: Done 

Editor’s comment: Discussion- 2nd para,6th line- 'unlike to this study'-delete 'to'

Response: Deleted 

Editor’s comment: 3rd para, 2nd line-'compared to our results which reflects the' delete 'which'

Response: Deleted 

Editor’s comment: 5th para,1st line-'On the other hand, previous treated patients'- write 'previously'

Response: Done 

Editor’s comment: 7th para, 2nd line-write geography instead of geographical

Response: Done 

Editor’s comment: 'This study; however, included data from 2016 and 2019'- replace 'and' by 'to', delete semi colon

Response: Replaced and deleted 

Editor’s comment: 'Xpert-MTB/RIF Assay is recommended to all TB suspected patients'- replace 'is' by 'was'. Also, delete 'with less likelihood of having RR-MTB as the MDR-TB suspected patients used in the other studies'.

Response: Replaced and deleted

---

## [Editor Report · Decision Letter 3]

23 Sep 2020

PONE-D-20-09432R3

Frequency of MTB and Rifampicin Resistance MTB using Xpert-MTB/RIF Assay among Adult Presumptive Tuberculosis Patients in Tigray, Northern Ethiopia: a cross sectional study

PLOS ONE

Dear Dr. Wasihun,

Thank you for submitting your manuscript to PLOS ONE. After careful consideration, we feel that it has merit but does not fully meet PLOS ONE’s publication criteria as it currently stands. Therefore, we invite you to submit a revised version of the manuscript that addresses the points raised during the review process.

The manuscript has been thoroughly revised. However there are a few minor corrections

Introduction: 1st para, 2nd line- Write Mycobacterium tuberculosis instead of mycobacterium tuberculosis

Inclusion criteria: indeterminate instead of intermediate (Had been pointed out earlier)

Discussion: 2nd para 6th line- delete 'to' in unlike to this study.(Had been pointed out earlier)

7th para, last line - 'MTB/RIF Assay was recommended to all TB suspected patients' replace 'to' by 'for'

We look forward to receiving your revised manuscript.

Kind regards,

Shampa Anupurba, MD

Academic Editor

PLOS ONE

---

## [Author Response · Author response to Decision Letter 3]

24 Sep 2020

Point-by-point response letter to Editor’s comments

Dear Editor,

Warm greeting,

First and foremost, we thank for giving a chance to revise our paper. We really appreciate the interest of the editor to our paper, and we also appreciate for the critical and important comments and recommendations. We have revised the manuscript and editor’s comments in highlighted in the revised manuscript. Finally, sorry for not addressing the points pointed out last time. 

Kind regards,

Araya Gebreyesus Wasihun (PhD)

 Editor’s comment: 

The manuscript has been thoroughly revised. However there are a few minor corrections

Editor’s comment: Introduction: 1st para, 2nd line- Write Mycobacterium tuberculosis instead of mycobacterium tuberculosis

Response: Corrected 

Editor’s comment: Inclusion criteria: indeterminate instead of intermediate (Had been pointed out earlier) 

Response: Corrected

Discussion: 

Editor’s comment: 2nd para 6th line- delete 'to' in unlike to this study.(Had been pointed out earlier)

Response: Deleted 

Editor’s comment: 7th para, last line - 'MTB/RIF Assay was recommended to all TB suspected patients' replace 'to' by 'for'

Response: Replaced

---

## [Editor Report · Decision Letter 4]

25 Sep 2020

Frequency of MTB and Rifampicin Resistance MTB using Xpert-MTB/RIF Assay among Adult Presumptive Tuberculosis Patients in Tigray, Northern Ethiopia: a cross sectional study

PONE-D-20-09432R4

Dear Dr. Wasihun,

We’re pleased to inform you that your manuscript has been judged scientifically suitable for publication and will be formally accepted for publication once it meets all outstanding technical requirements.

Kind regards,

Shampa Anupurba, MD

Academic Editor

PLOS ONE
---

## [Editor Report · Acceptance letter]

30 Sep 2020

PONE-D-20-09432R4 

Frequency of MTB and Rifampicin Resistance MTB using Xpert-MTB/RIF Assay among Adult Presumptive Tuberculosis Patients in Tigray, Northern Ethiopia: a cross sectional study 

Dear Dr. Wasihun:

I'm pleased to inform you that your manuscript has been deemed suitable for publication in PLOS ONE. Congratulations! Your manuscript is now with our production department. 

Kind regards, 

on behalf of

Dr. Shampa Anupurba 

Academic Editor

PLOS ONE